# Efficient Retention and Alpha Spectroscopy of Actinides from Aqueous Solutions Using a Combination of Water-Soluble Star-like Polymers and Ultrafiltration Membranes

**DOI:** 10.3390/polym14173441

**Published:** 2022-08-23

**Authors:** Valery N. Bliznyuk, Nataliya V. Kutsevol, Yuliia I. Kuziv, Scott M. Husson, Timothy A. DeVol

**Affiliations:** 1Environmental Engineering and Earth Sciences, Clemson University, Anderson, SC 29625, USA; 2Faculty of Chemistry, Taras Shevchenko National University of Kyiv, 60 Volodymyrska Street, 01601 Kyiv, Ukraine; 3Institut Charles Sadron, Universite de Strasbourg, CNRS UPR22, 23 Rue du Loess, CEDEX 2, 67034 Strasbourg, France; 4Chemical and Biomolecular Engineering, Clemson University, Clemson, SC 29634, USA

**Keywords:** extractive membranes, polymer-enhanced ultrafiltration, water-soluble polyacrylamides, uranium extraction, plutonium extraction

## Abstract

We explored two approaches to recover uranium and plutonium from aqueous solutions at pH 4 and pH 7 using water-soluble star-like polyacrylamide polymers with a dextran core. In the first approach, a solution comprising a neutral or ionomer polymer was mixed with a radionuclide solution to form polymer–metal complexes that were then retained by ultrafiltration (UF) membranes under applied pressure. The same polymers were first deposited on the membrane in the second approach using pressure-driven flow. The applied polymers had an overall diameter of gyration of 120 nm, which exceeded the nominal diameter of the UF membrane pores. The polymers showed a high affinity to uranyl but could also be used to extract Pu from neutral or near-neutral pH solutions. Direct-flow single-step filtration and alpha spectrometry demonstrated that the UF membranes containing star-like copolymers could recover 99% of U and up to 60% of Pu from deionized water after filtering 15 mL solutions containing 25 ppm and 33 ppb of the actinides, correspondingly. The sorption capacity of the polymers for uranium could be measured as 1mg U per mg of the polymer after six subsequent filtration steps. Alpha spectroscopy of the deposited actinides revealed peculiarities of the structural organization of polymers and their complexes with U or Pu, depending on the approach. Though both approaches were efficient, the second approach (deposition of the polymer on the membrane followed by filtration) has an additional advantage of protecting the membrane pores from capillary collapse by filling them with the polymer chains. Therefore, these polymer-modified membranes could be used either in continuous or multi-step filtration process with drying after each step without deterioration of their sorption characteristics.

## 1. Introduction

Purifying water contaminated with heavy metals or radionuclides became a critical issue with industrial development and various human activities [1]. Developing efficient yet inexpensive purification techniques for actinides is crucial for many situations associated with nuclear energy [2] and the concentration of U from seawater [3]. In addition, quantifying water contamination with some radioisotopes (including uranium) is necessary for consuming natural groundwater, nuclear power plant accidents, leakage from nuclear waste storage, nuclear forensics and reduction of nuclear terrorism threads.

Previously, we demonstrated how a combination of water-soluble star-like polyacrylamides and kaolin clay minerals can be used for fast and simple sedimentation of uranium, cesium and strontium via flocculation [4]. The efficiency of the process was due to the complexation of functional groups of the polymer with metal cations. Generally, the interaction of various polymer sorbents with heavy metals, particularly uranium, was a subject of intensive studies during the last two decades [5]. The best polymers, such as polyamidoxime and polydopamine and hybrid organic/inorganic systems (graphene oxide, organic metal frameworks, and zeolites) show high sorption capacity and selectivity to uranium [6]. As a separate approach, functionalization of polymer membranes or flat Teflon plates with U- or Pu-selective ligands such as polyamidoxime or poly(ethylene glycol methacrylate phosphate), enabling express alpha spectroscopy detection of such radionuclides was also reported [7,8,9]. Most of the suggested materials are expensive or have relatively slow uptake kinetics and therefore need further development before their upscaling from the lab to industry applications. Nevertheless, the performed studies allowed for the formulation of structure–property relationships for the rational design of efficient sorbents. Thus, high affinity and selectivity to uranium can be achieved when a combination of oxygen-containing and amino groups of some particular 3D arrangement (position and orientation) is used (e.g., in amidoxime groups) [6]. Such architecture of the functional groups is dictated by the shape and functionality of uranyl ion (UO_2_^2+^), the main structural unit of water-soluble uranium salts. 

Since the macromolecular architecture plays an important role in the design of polymer sorbents, we have focused our studies on the application of polymers with special supramolecular structures for retention of radionuclides using ultrafiltration (UF). Brush-like and star-like polyacrylamides (PAAm) have been reported as being effective materials for removing radionuclides from water [10,11,12,13,14]. In this study, several dextran-*graft*-PAAm copolymers with varied dextran core sizes and conformational state of the PAAm grafts have been applied in combination with UF membranes to achieve fast and efficient recovery of U and Pu from water solution. 

Combining water-soluble polymer sorbents with ultrafiltration is a well-studied process known as polymer-enhanced ultrafiltration (PEUF) [15]. The method involves the addition of functional macromolecules to the aqueous solution with contaminants before filtration. Due to the interaction between the macromolecules and organic contaminants or metal ions dissolved in water, the latter can be removed from the solution during the subsequent UF process. For PEUF, the size of the macromolecules has to be larger than the so-called molecular weight cut-off (MWCO) of the UF membrane. PEUF works well in many situations but has drawbacks of increased hydraulic resistance of the membrane (resulting in reduced water flux), possible secondary contamination of the solute, as well a more complicated technological scheme leading to increased purification cost [16,17,18,19,20]. However, these drawbacks are unimportant for the current application that focuses on uranium retention for detection by alpha spectroscopy.

There are two general classes of water-soluble polymers for PEUF: polychelatogens and polyelectrolytes. Polychelatogens are macromolecules containing chelating or complexing groups, while polyelectrolytes dissociate in polar solvents into charged polymer chains (macro-ions) and small counter ions [21]. Both polymers can interact and retain metallic cations in solution due to their ion exchange or complexing properties. The functional groups on the polymer macromolecules act as sites to sorb heavy metals or small organic matters via coordination or Coulombic interactions. 

The water-soluble star-like PAAm used in this study has the features of both polymer classes. Being originally synthesized as neutral polymers with chelating acrylamide groups, these macromolecules can be converted into polyelectrolytes via polymer–analogous transformation (saponification) with up to 30% of COO^−^ ionic groups. Moreover, the macromolecular architecture enables two different variants of PEUF: a traditional PEUF described in the literature and a new one (not reported before) involving trapping the star-like macromolecules inside the pores of a UF membrane. 

Therefore, the objective of this study was to determine the effectiveness of star-like polyacrylamide-based copolymers for actinide retention and isotopic analysis using complexation-filtration and filtration through polymer-modified ultrafiltration membranes. The characteristic features of the metal ion-polymer interaction process and conformational changes of the macromolecules during filtration have been revealed by a combination of spectroscopy (alpha spectroscopy and FTIR) and scanning electron microscopy (SEM) techniques.

## 2. Materials and Methods

### 2.1. Materials

In uranium adsorption experiments, 25 ppm solutions ^238^U in deionized (DI) water spiked with 6 Bq/mL ^233^U at pH 4 or 7 were used. Such solutions were prepared by dilution and pH adjustment (addition of NaOH) of a 75-ppm stock solution at pH 3 just before the experiments to avoid aggregation/precipitation of uranium.

Diluted ^242^Pu solutions were prepared using a 147 Bq/mL ^242^Pu(VI) stock solution in 1 mM HNO_3_ provided by Dr. Brian Powell (Clemson University Department of Environmental Engineering and Earth Sciences). Mass fractions of the Pu isotopes in the stock solution are provided in Appendix A. Appendix A shows the solution’s corresponding 1 h alpha spectrum.

Synder A6 membranes (0.0635 mm PVDF top layer, 50 nm average pore diameter, 500 kDa MWCO) were purchased in 305 × 305 mm^2^ flat sheets from Sterlitech (Kent, WA, USA). Ultima Gold liquid scintillation cocktail was purchased from PerkinElmer (Waltham, MA, USA). Nitric acid (HNO_3_, 65% *w*/*w*) and sodium hydroxide (NaOH, 97%) were purchased from Fisher Scientific (Waltham, MA, USA). 

### 2.2. Synthesis of Star-like Polyacrylamides

Water-soluble dextran-*graft*-polyacrylamide (DXX-PAAm) copolymers for this study were synthesized as previously described [22,23]. These copolymers consist of dextran core of various molecular mass (M_w_ = 20,000, 70,000, and 500,000 g/mol) and polyacrylamide grafts (see Appendix A for more details). The “XX” in the co-polymer naming convention refers to the molecular mass of the dextran core divided by 1000. Ceric ion-induced redox initiation was used for copolymer synthesis. The redox process initiates free radical sites exclusively on the polysaccharide backbone, thus preventing the formation of homopolymer polyacrylamide (PAAm) [22]. The mechanism of ceric ion initiation involves the formation of a chelate complex, which decomposes and generates free radical sites on the polysaccharide backbone. Free radicals trigger the growth of grafted chains in the presence of acrylic monomer. The suggested reaction path is shown in Figure 1. 

Synthesized copolymers are star-shaped with a dextran core and polyacrylamide grafts [22,23]. The dextran retains a macrocoil conformation during synthesis, as the Ce(IV)-ions promote the crosslinking of dextran macromolecules. The conformation of grafted PAAm chains depends on the distance between grafts (see Figure 1). For synthesized D20-PAAm, the grafted chains have worm-like conformation, whereas D70-PAAm and D500-PAAm have a mushroom conformation [23]. 

The copolymers were saponified according to Figure 2 shown below to obtain star-like polyelectrolytes. 

It is evident that branched polymer systems are more compact than their linear analogs; thus, they have a higher local concentration of functional groups than the linear polymers. Thus, such copolymers can be more efficient than sorbents in removing contaminants (clay particles, metal ions, etc.) from water [4].

### 2.3. U, Pu Retention Experiments

Figure 2 shows two approaches for recovering radionuclides from water using a combination of water-soluble polymer and UF membrane. In the first approach, the water-soluble polymer is added to an aqueous solution contaminated with uranium. This leads to the formation of metal–polymer complexes due to charge interaction between polymer chains and metal ions. The process may happen relatively fast (within seconds or minutes), depending on the concentrations of the components. In the second step of this approach, the formed metal–polymer complexes are deposited on the UF membrane surface when the solution is forced to pass through the membrane under the application of applied pressure. The pore size of the membrane is smaller than the size of the formed metal–polymer complexes, which assures the complete retention of the polymer.

In the second approach, the polymer is first deposited on the membrane surface from a water solution, followed by filtration of the uranium or plutonium solution. The first step causes the formation of a thin layer of the polymer that is permeable to water. In the second step of this approach the aqueous solution of radionuclides is placed in a feed reservoir in contact with the modified membrane for 15 min and then passed through the membrane by application of applied pressure. During this step, water permeation and metal–polymer complexes’ retention occur simultaneously. The second step can be repeated several times (multistep filtration) with or without drying the polymer film between steps leading to the accumulation of radionuclides in the polymer layer up to the limit of the capacity of the polymer sorbent. An ancillary benefit of the second approach is that it protects pores from collapse under capillary forces caused by drying. During polymer deposition, polymer chains partially penetrate inside the pores. Thus, the membranes can be stored dry without concern for the loss of permeability due to pore collapse.

In both approaches, Synder PVDF A6 membranes were placed in a 45 mm diameter Amicon ultrafiltration cell. The membrane samples were fastened to the bottom of the flow cell, and the cell was filled with 15 mL of polymer–actinide solution of a desirable composition, concentration, and pH (Approach 1). The cell was sealed, left at atmospheric pressure for 15 min and then pressurized to 100–140 kPa. The permeate was collected for activity measurements. The membrane was removed once the cell was empty and depressurized. The membrane was left to dry for at least 3 h prior to alpha spectrometry measurements. 

In Approach 2, the same ultrafiltration cell was used first to deposit polymer from water solution on the surface of the Synder PVDF A6 membrane and then for passing an actinide solution as described above. Gross activity measurements were made on the collected permeate solutions after combining with a scintillation cocktail and counted using the Quantulus LSC. Comparison with the activity of the original (control) solution gave the percentage of retained radionuclides by the equation: R% = [1 − (A_C_ − A_S_)/A_C_] × 100%. Where A_C_ and A_S_ are activities of the solution before and after filtration.

### 2.4. Alpha Spectrometry Measurements

Alpha spectrometry measurements of radionuclide-loaded membrane samples were performed using a Canberra (Mirion) 7401 alpha spectrometer (Meriden, CT, USA) with a 450 mm^2^ passivated ion silicon (PIPS) detector. Alpha peak energy and efficiency calibration was performed using a NIST-traceable Eckert & Ziegler electrodeposited source containing known activities of ^234^U, ^238^U, ^239^Pu, and ^241^Am on a 30 mm diameter steel planchette.

After filtration of the uranium or plutonium solutions, UF membranes were placed on a planchet and held flat using a 3-D printed plastic fixture. The membrane surface was positioned 5 mm from the PIPS detector and counted for 1 h. Ortec (Ametek) APEC-927 (Oak Ridge, TN, USA) multichannel analyzer coupled with MAESTRO Ver. 7.01 software program (AMETEK Inc., Berwyn, PA, USA) to collect, display and analyze the alpha spectra.

### 2.5. SEM Analysis of Membrane Morphologies

Scanning electron microscopy imaging was used to visualize morphology and locate where star-like polymer resides within the membrane structure. The pristine membrane was prepared for electron microscopy by the procedure described in [24]. In short, the pores were first washed with DI water to remove any humectant from the manufacturer and then the water was replaced with ethanol and hexamethyldisilazane (HMDS) by immersing the membrane in solutions of water/ethanol and then ethanol/HMDS with concentration changing from 100% of the first component to 100% of the second one. Samples were dried and mounted to aluminum holders with carbon tape and then coated with platinum for 2 min using an Anatech Hummer 6.5 sputter coater (Sparks, NV, USA). The Pt coating thickness was 4 nm. Micrographs of the membrane surfaces were taken using a Hitachi Regulus 8230 SEM (Tokyo, Japan) at an accelerating voltage of 20 kV. 

### 2.6. Infrared Spectroscopy

Attenuated total reflection mode Fourier transform Infra-Red spectroscopy (ATR-FTIR) was applied to study the chemical composition of UF membranes after modification with star-like polymers and the formation of polymer–uranium complexes. A piece of UF membrane was placed directly onto a Smart iTR single-bounce diamond ATR crystal and analyzed using a Thermo Scientific 6700 FTIR equipped with a mercury cadmium telluride narrow band detector.

## 3. Results and Discussion

Table 1 summarizes uranium uptake values for uranium retention from water solution depending on the applied water-soluble polymer and approach. The uptake of a bare A6 UF membrane is shown for comparison. In both approaches, the interaction time of polymer molecules with uranyl ions was kept the same (~15 min). As can be seen, higher retention (close to 100%) can be achieved in Approach 2 when the polymer film was first deposited on the membrane and then used for filtration of the U solution. A slightly lower uptake of U in Approach 1 probably indicates that the interaction time was not sufficient for the formation of 100% polymer–metal complexes, and the process is diffusion controlled. There was no difference among the polymers with the different molecular mass of the dextran core in retention properties for the neutral polymers; however, the ionic polymers generally display a bit lower retention under the same condition and reduced uptake for the higher molecular mass core. The effect can be understood by comparing the PAAm shell’s conformational state depending on the core size (an extended worm-like conformation of the PAAm arms in D20-PAAc polymer is replaced with a mushroom-like polymer for D70-PAAm and D500-PAAm). Again, a more compact conformation of the arms in the latter case requires a longer time for untangling and complexation with uranyl ions.

Figure 3 shows the alpha spectra of the samples after deposition on PVDF A6 membrane and drying. The spectra contain two characteristic peaks corresponding to the main uranium isotope (^238^U) and a tracer (^233^U). The latter peak is much stronger due to the 10^3^ times higher activity of the isotope. Alpha spectroscopy confirms a high concentration of U deposited on a UF membrane under the application of each of the two approaches.

Moreover, Figure 3 reveals a remarkable difference in the shape of alpha spectra of polymer–uranium complexes formed in Approach 1 and Approach 2. Particularly, for the same amount of deposited polymer and the same amount of adsorbed uranium, the peaks are much broader and are shifted to lower energy for Approach 2 deposition compared with the corresponding peaks for Approach 1 (complexation/filtration). The same trend was observed for all polymers under study regardless of their ionic state and the size of the dextran core. Figure 2 also shows the alpha spectrum of a control sample, which was obtained by depositing uranium on a bare PVDF A6 membrane without star-like PAAm. The retention ability of the nonmodified membrane is much lower. Therefore, the intensity of the peaks is smaller (in fact, only the peak corresponding to a higher activity isotope ^233^U is detected in this case). The ^233^U alpha peak of the control sample has the highest energy (810 channel #) compared with ~800 Ch# in the Approach 1 sample and 745 Ch# in the Approach 2 sample. 

The observed difference in the energy of the peaks corresponds to some attenuation (shielding) of alpha particles emitted by U atoms in the polymer film (especially in Approach 2) in comparison with the bare membrane where one can expect the location of the uranium atoms on the top surface of the membrane. The significant difference between Approach 1 and 2 spectra suggests that in the latter case, uranyl ions can penetrate deep inside the membrane while sitting mainly within the star polymer film after the Approach 1 deposition.

To shed more light on this different behavior, we performed additional experiments with a variation of the mass of polymer used for uranium retention. The results are presented in Figure 4. A decrease in the mass of polymer sorbent leads to the lower intensity of the ^233^U peak as well as to its shift to higher energy (Approach 1). Dramatic changes were observed for the alpha spectra of uranium in the case of Approach 2. Here, the reduction in the mass of the deposited polymer leads to an abrupt sharpening of the peak, an increase in intensity and a shift of the peak maximum to higher energy. This variation of the spectrum is clearly seen within the range of 0.2–1.0 mg of D500-PAAc polymer per 45 mm diameter membrane. Further reduction of polymer mass leads to a reduction in the peak intensity. Calculations predict that ~0.5 mg of the polymer corresponds to forming a complete monomolecular coating of the membrane with the star-like polymer. With a lower mass of the polymer, incomplete coverage can be expected, and, therefore, the polymer should form some islands on the surface of the membrane. 

The plutonium uptake experiments are summarized in Figure 5. The data demonstrate a much lower uptake of Pu in comparison with U. Analysis of corresponding alpha spectra revealed the presence of two peaks corresponding to ^242^Pu and ^241^Am alpha-emitting isotopes (Figure 6). The latter component appears due to the nuclear decay chain of the ^241^Pu isotope, which is present in a trace amount in the ^242^Pu stock (see Appendix A). The activity of ^241^Am can be calculated based on the original concentration of ^241^Pu, the known half-life time parameter and the time lapsed from the stock solution preparation. The corresponding calculation reveals that the activity of ^241^Am is approximately 50% of the activity of ^242^Pu for the time when retention experiments were carried out.

Interestingly, the ratio of the two peak intensities I_Pu-242_/I_Am-241_ varies with the size of the polymer dextran core and is very sensitive to the type of the PAA shell (ionic versus neutral) but appeared to be insensitive to the deposition approach. Thus, this ratio is reduced from almost 2 (D70-PAAc) to around 0.5 (D70-PAAm and D500-PAAm neutral forms). Generally, the ratio is above 1 for all polymers with ionic PAAm chains and below 1 for all neutral polymers (see Appendix A and Appendix A). The I_Pu-242_/I_Am-241_ ratio can be as high as 2 if both Pu and Am have equal affinity to the polymer chains. Moreover, the higher energy peak in Figure 5 may have an additional contribution from alpha decay of ^238^Pu isotope (E_α_ = 5.499 MeV), which is present in the stock solution as a trace impurity and may appear as a shoulder in the alpha peak of ^241^Am. 

Therefore, at least on a qualitative level, one can see that for the ionic PAAc chains, the interaction of the polymer with Pu and Am ions is about the same. On the contrary, in the case of the neutral polymer form (especially D500-PAAm and D70-PAAm), competition between Am and Pu ions for functional sites on the polymer is a critical factor. It is in favor of the Am-polymer complexation.

The IR spectra of the modified membranes show characteristic vibration bands corresponding to PVDF (1400, 875, and 840 cm^−1^) as well as the ones that belong to polyacrylamide polymer (mainly in the range from 1750 to1450 cm^−1^) (Figure 7). The ionic modification of PAAm indicates the presence of carboxylic groups with three strong bands at 1717, 1541, and 1456 cm^−1^ corresponding to ν(COOH) and ν_s_(COO^−^)/ν_as_(COO^−^), respectively, which do not appear in the neutral form of the same polymer [25]. The neutral form of the polymer PAAm displays characteristic peaks at 1656 and 1612 cm^−1^ corresponding to -CO-NH_2_ groups in the PAAm shell. The adsorption of uranyl is manifested by characteristic antisymmetric stretching band ν_as_(UO_2_^2+^) in the range 960–900 cm^−1^ [26]. The formation of complexes of uranyl with polymer provokes a strong bathochromic shift of the asymmetric mode. A frequency of around 920 cm^−1^ is characteristic of the complexation of uranyl with carboxylate groups [27] and depends strongly on the strength of the interaction. The position of this band is different for ionic and neutral forms (D70-PAAm and D70-PAAc polymers are shown as an example in Figure 7), which indicates the difference in the strength of interaction of the uranyl units with the polymer in corresponding chelate complexes.

Peculiarities in the morphology of modified UF membranes and adsorbed polymer–uranium complexes were revealed by SEM studies (Figure 8). The pristine membrane surface shows a number of pores with a diameter below 100 nm, as well as agglomerates of such pores of larger size (black spots in the image). The same pores can be observed for membranes with deposited polymer but only in the case of low polymer concentrations. Calculations based on the known size of polymer particles predict that 0.5 mg of the polymer is just enough to make complete coverage of the membrane surface (45 mm diameter) with monomolecular layer.

These images demonstrate the influence of ionic groups on the morphology of polymer films deposited on the A6 membrane. D20-PAAm polymer possesses a granular surface with a characteristic size of the granules closely matching the size of macromolecules. On the contrary, extremely smooth featureless morphology can be seen for the ionic form of the same polymer. The difference may be attributed to the difference in drying conditions for the two polymers. The ionic form of the polymer is highly hydrophilic and, therefore, can retain water for a longer time, thus reducing the drying rate. As a result, polymer chains have a longer time for conformational changes and establishing intermolecular interactions, which reduces the chances for disentanglement of macromolecular chains of neighboring molecules. Ionic polymers generally show smoother surface topography in comparison to their neutral analogs. The smoother surface is also observed for higher molecular weight polymers (D500 in comparison to D20).

Comparing the two approaches for polymer–metal complexes formation also reveals some interesting behavior. Approach 1 proceeds through forming polymer–U complexes in the solution state and bridging between macromolecules by uranyl units. This leads to smoother surface morphology after deposition of the complexes on the UF membrane during filtration. In Approach 2 polymer–metal complexation happens on the polymer film interface. During such interaction, some segments of the polymer chains expand towards the water solution and then shrink to a more compact state during filtration and drying. Therefore, rougher surface morphology develops after several deposition cycles. The image in Figure 4 corresponding to three consequent depositions of uranium from an aqueous solution on a deposited polymer film reveals such behavior. Alpha spectra of such membranes after drying (Appendix A) display strong ^233^U and ^238^U peaks corresponding to a high accumulation of uranium (~1 mg per mg of the active component D500-PAAc). Notably, this case also demonstrates the possibility of application of the same modified membrane several times with drying after each filtration step, which would not be possible without star-like polymer deposition.

Nonionic polymers have yellowish color after deposition on the membranes, distinguishing them from the snow-white appearance of the ionic analogs with the same other parameters (polymer concentration, deposition pressure, etc.). Electron microscopy revealed supramolecular organization in such films after deposition (Appendix A). Regular density variation appears as a result of self-organization during the filtration process. We can assume that the color of the films is a structural color due to nanoscale modulation of the density and, therefore, of the refractive index within the film.

Based on SEM images and alpha spectra of the modified membranes, we theorize that the conformation of the DXX-PAAm macromolecules changes to a “flat”, in-plane structure on the membrane surface during the applied deposition process. Due to a strong dragging force of the water flow through the pores of the UF membrane, the arms of star-like polymers are forced to expand and penetrate inside the pores, thus partly blocking them. After the forced deposition under pressure, these polymer chains serve as anchors making desorption unfavorable and simultaneously protecting the membrane pores from capillary collapse (i.e., the collapse of the pores due to an action of capillary forces of water). We believe this situation is unique for star-like polymers with a proper architecture (i.e., the overall diameter and the length of the side chains). Linear polymer chains have the chance to elongate and be transmitted through the pores even when their molecular mass is above the membrane molecular weight cut-off limit. Dendritic molecules do not have enough mechanical flexibility to be deformed to a high extent [28,29] and can be desorbed afterward under appropriate conditions (pH and charges in the solution). In the case of linear or dendritic macromolecules, the pores of the membrane remain unfilled and therefore collapse after the filtration process if the membrane is removed from solution and dried.

## 4. Conclusions

The ability of branched star-like polyacrylamides to form stable complexes with uranyl in water has been confirmed. Such water-soluble polymers can be applied in combination with ultrafiltration membranes to retain up to 1 g of uranium per gram of polymer from neutral water solutions. Our suggested approach combines novel polymers with filtration methods that enable the rapid and selective concentration and direct detection of waterborne radioactive debris.

The star-like polymers can be mixed with uranium solution and then filtered out through a membrane (complexation/filtration protocol) or can be used for the formation of a nanoscale coating on a membrane that can be used further for retention of uranium (deposition/filtration protocol). Similar experiments with Pu demonstrated less efficient retention of the element. The most efficient extraction of actinides in both cases is attained with the star-like polymer with high molecular mass dextran core and PAAm arms with a large number of ionic carboxylate groups in the composition. UF membranes with such polymers deposited on the surface by pressure cells can be dried and stored and/or used several times to retain U or Pu at near-neutral conditions without deterioration of membrane permeability. The modified membranes have good potential for fast in situ purifications of water from actinides. Moreover, the second approach of UF modification via partly blocking the pores with water-soluble star-like functional polymers represents a new technique of membrane modification suitable for the design of membrane/polymer systems with high efficiency and specificity to particular organic or inorganic contaminants.

## Data Availability

Not applicable.

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
