# Peer review of "Efficient Retention and Alpha Spectroscopy of Actinides from Aqueous Solutions Using a Combination of Water-Soluble Star-like Polymers and Ultrafiltration Membranes"

_polymers, 2022, doi:10.3390/polym14173441_

Round 1

Reviewer 1 Report

Manuscript Title: Efficient retention and alpha spectroscopy of actinides from aqueous solutions using a combination of water-soluble star-like polymers and ultrafiltration membranes

Journal Title: Polymers

Authors: Valery N. Bliznyuk, Nataliya V. Kutsevol, Yuliia I. Kuziv, Scott M. Husson, Timothy A. DeVol

Manuscript ID: polymers-1838043

The work is sutable for publication in Polymers taking into account the novelty, the direction of the research and the quality of the presentation of scientific material and discussion. However, this version of manuscript is not ready for publication in present form, there are major concerns to be addressed before its publication as follows:

- Abstract should specified with more details.

- Materials and methods. “Mass fractions of the Pu isotopes in the stock solution are provided in Table S1 of Supporting Information”. Please check, I guess it should be Table S2. 

- The choice of the ultrafiltration membrane with certain parameters should be justified and discussed.

- Please check time of platinum sputtering “coated with platinum for 40 min using an Anatech Hummer 6.5 sputter coater”. Seems like a very long time. What was a thickness of the platinum layer on the membrane?

- Results and Discussion. “The uranium solution was kept on top of membrane for 15 min before application of the external pressure in each case”. Please discuss why it should be done.

- Figure 5. Please correct X-axis for printed matter. If possible, please add error bars.

- Since this work is also of practical importance, the authors are recommended to add a comparative table for extracting of uranium and Pu.

- The authors are recommended to study adsorption mechanism (adsorption isotherm) and kinetics. They are also important for understanding the processes.

- References, out of 29 references 8 references belong to authors (almost 30%) are self-citations. It is recommended to reduce number of self-citations. 

Author Response

Dear Editor,

We are deeply grateful to you and the referees for the feedback and consideration of our work. In our revised manuscript, we have tried to address the reviewers’ concerns and answer their questions. Please see below our point-by-point answers and explanations to changes that were made in the text.

Reviewer 1

- Abstract should specified with more details.

More details have been added to the Abstract.

- Materials and methods. “Mass fractions of the Pu isotopes in the stock solution are provided in Table S1 of Supporting Information”. Please check, I guess it should be Table S2. 

We have changed the order of Tables S1 and S2 to avoid this confusion.

- The choice of the ultrafiltration membrane with certain parameters should be justified and discussed.

- Please check time of platinum sputtering “coated with platinum for 40 min using an Anatech Hummer 6.5 sputter coater”. Seems like a very long time. What was a thickness of the platinum layer on the membrane?

We thank the reviewer for finding this typo. The actual sputtering time was 2 min, which resulted in ~ 4 nm thick metal film. We have added the conductive layer thickness to the Experimental Section.

- Results and Discussion. “The uranium solution was kept on top of membrane for 15 min before application of the external pressure in each case”. Please discuss why it should be done.

This time was chosen based on our previous experiments with various polymer ligands. Complexation of U or Pu with polymer ligand requires some time for completion. Very high uptake values for uranium give evidence that 15 min was long enough for the system under study.

- Figure 5. Please correct X-axis for printed matter. If possible, please add error bars.

Figure 5 has been corrected and error bars were included.

- Since this work is also of practical importance, the authors are recommended to add a comparative table for extracting of uranium and Pu.

Direct comparison between U and Pu adsorption is hardly possible due to large difference in their concentrations. This difference is due to high specific activity and lower solubility limit (at pH7) of Pu in comparison with U.

- The authors are recommended to study adsorption mechanism (adsorption isotherm) and kinetics. They are also important for understanding the processes.

We agree. However, measurements of adsorption isotherms and kinetics of the process were outside the scope of funded work and will need to be subject of separate study.

- References, out of 29 references 8 references belong to authors (almost 30%) are self-citations. It is recommended to reduce number of self-citations. 

In accordance to this suggestion, we have reduced the percentage of self-citations to 24%.

Reviewer 2 Report

Major Revision

The author has demonstrated the feasibility of polymer enhanced ultrafiltration membrane application to recover uranium and plutonium from aqueous solutions. Overall, this is an interesting research topic which can be noteworthy for researchers’ studying membrane and recovering uranium and plutonium. However, in this research paper, there are few important queries which should be addressed before publication.

1.     Starting with the introduction part, the state-of-art can be added by comparing the previous research outputs (conditions/parameter/overall results) with the present one in tabular form to show the viability of the present study.

2.     The objective of this research article is missing. It has to be included in the introduction section for better understanding (specifically last paragraph).

3.     Why only recovery of uranium and plutonium has been demonstrated? Please specify the reasons.

4.     How this kind of research approach had the potential to be a solid contribution in the field polymer enhanced ultrafiltration? Kindly comment on this.

5.     While dealing with UF membranes, the author must indicate following results which is totally missing from the manuscript (if possible):

(1)   Pore size distribution/Porosity of various membranes

(2)   Thickness of membranes (since it may affect the mass transfer).

Author Response

Dear Editor,

We are deeply grateful to you and the referees for the feedback and consideration of our work. In our revised manuscript, we have tried to address the reviewers’ concerns and answer their questions. Please see below our point-by-point answers and explanations to changes that were made in the text.

Reviewer 2

  1. Starting with the introduction part, the state-of-art can be added by comparing the previous research outputs (conditions/parameter/overall results) with the present one in tabular form to show the viability of the present study.

There are hundreds of recent publications considering various polymer, organic or hybrid ligands for selective extraction of actinides from aqueous solutions. Moreover, different authors are using dozens of experimental techniques to study extraction. We believe that deep insight and comparison among different retention/filtration/sensing techniques previously developed for uranium and plutonium would go far beyond the scope of our study and would be more suitable for a review paper. The scope of our study is different and includes extraction of U and Pu combined with fast preparation of samples for alpha spectroscopy, which allows identification of isotopic composition. We modified several sentences in the Introduction to clarify this point and added 3 relevant recent references.

  1. The objective of this research article is missing. It has to be included in the introduction section for better understanding (specifically last paragraph).

We have modified the Introduction in accordance with this suggestion.

The objective of this study was to determine the effectiveness of star-like polyacrylamide-based copolymers for actinide retention and isotopic analysis using complexation-filtration and filtration through polymer-modified ultrafiltration membranes.

  1. Why only recovery of uranium and plutonium has been demonstrated? Please specify the reasons.

Uranium and plutonium are considered to be the most important actinides used in nuclear energy generation and nuclear weapons production. Therefore, we believe they are the key elements in estimation of risks associated with nuclear fuel production, nuclear waste storage and deactivation as well as in homeland security and nuclear forensics.

  1. How this kind of research approach had the potential to be a solid contribution in the field polymer enhanced ultrafiltration? Kindly comment on this.

Our suggested approach combines novel polymers with filtration methods that enable the rapid and selective concentration and direct detection of waterborne radioactive debris.

Therefore, our method can potentially 1) simplify UF membrane functionalization process, 2) reduce secondary contamination of water associated with filtration, 3) speed up the detection of actinides and determination of their isotopic composition, 4) reduce associated costs and enable portative in field studies of the type of radionuclides present in water for the purpose of water purification or nuclear forensics.

We have modified slightly the Conclusions to include this summary.

  1. While dealing with UF membranes, the author must indicate following results which is totally missing from the manuscript (if possible):

(1)   Pore size distribution/Porosity of various membranes

(2)   Thickness of membranes (since it may affect the mass transfer).

The manufacturer has following information on PVDF A6 membranes: average pore diameter 50 nm, thickness of the active PVDF layer 0.0635mm, thickness of backing polyester film 0.1524mm.

The pore size distribution was not determined, but as can be seen from our SEM images this distribution is rather broad. The most important parameter is the molecular weight cut off (MWCO), since that influences polymer retention. This parameter is equal to 500 kDa in accordance to the manufacturer specification.

In accordance with this suggestion of the reviewer, we have added the most important parameters of the A6 UF membrane to the Experimental section.

Round 2

Reviewer 1 Report

The authors have addressed all my comments/concerns in this revised manuscript. I have no additional concerns regarding this manuscript.

Manuscript could be recommended for publication in this form

Reviewer 2 Report

The authors have addressed all my queries well and satisfactorily with high scientific discussions. Therefore, the revised article can be accepted in the present format.